# The Protection Role of Cysteine for Cu-5Zn-5Al-1Sn Alloy Corrosion in 3.5 wt.% NaCl Solution

**Kebede W. Shinato †, Feifei Huang †, Yanpeng Xue, Lei Wen and Ying Jin ***

National Center for Materials Service Safety, University of Science and Technology Beijing, Beijing 100083, China

* Correspondence: yjin@ustb.edu.cn; Tel.: +86-17-8010-50028

† These two authors contribute equally to this manuscript.

**Abstract:** In this work, the corrosion mechanism of a Cu-5Zn-5Al-1Sn alloy was examined in a 3.5 wt.% NaCl solution. At the same time, the effect of a cysteine inhibitor was also investigated through a multi-analytical approach. Electrochemical results suggested that inhibition efficiency increased with the increase of cysteine concentration. From potentiodynamic polarization (PD) analysis, a decrease in corrosion current and corrosion potential shift toward a more negative direction was observed. The potential difference between the blank and inhibited surface was found to be 46 mV, which is less than 85 mV, revealing a mixed type inhibition effect of cysteine for the Cu-5Zn-5Al-1Sn alloy. The inhibition mechanism of cysteine (Cys) and the effect of alloying elements were investigated by fitting experimental impedance data according to a projected equivalent circuit for the alloy/electrolyte interface. A Langmuir adsorption isotherm was proposed to explain the inhibition phenomenon of cysteine on the Cu-5Zn-5Al-1Sn alloy surface. Surface morphology observation confirmed that the Cu-5Zn-5Al-1Sn alloy was damaged in 3.5 wt.% NaCl solution and could be inhibited by using the cysteine inhibitor. The impact of alloying elements on the corrosion mechanism was further examined by surface analysis techniques such as X-Ray photoelectron spectroscopy (XPS)/Auger spectra, the results of which indicated that the corrosion inhibition was realized by the adsorption of the inhibitor molecules at the alloy/solution interface.

**Keywords:** corrosion inhibitor; corrosion mechanism; cysteine; thin film

## 1. Introduction

In recent years, research on copper alloy corrosion have garnered more attention in the industrial sector since copper and its alloys form a number of industrially important materials [1]. Based on alloying elements, the important properties range from good thermal properties, electrical conductivities, formability, and visual appearance, to good corrosion resistance [2–6]. Cu-5Zn-5Al-1Sn (Cu5Zn5Al1Sn) consists of 89 wt.% copper, 5 wt.% aluminum, 5 wt.% zinc, and 1 wt.% tin, with the latter usually used for the front covering due to its shiny and golden appearance [7]. The features of the microstructure of the Cu5Zn5Al1Sn alloy have been studied by comparing it with pure copper [8]. The microstructure of Cu5Zn5Al1Sn owns numerous properties that are promising from a corrosion resistance perspective. These properties consist of meaningfully smaller grain size (2.1 μm versus 10.4 μm) and a higher segment of coherent twin boundaries [8]. Smaller grain size and a closer, well-ordered grain boundary illustrate the relatively high corrosion resistance of the alloy [8–10]. The Cu5Zn5Al1Sn alloy forms relatively poorly soluble oxide films with its constituent metals in the presence of chlorides when exposed to long-term outdoor and short-term laboratory conditions [11]. Even though corrosion resistant behavior of this golden alloy is much better than that of other comparable alloys, it is found to be easily corroded in acidic media. Similarly to Cu metal, $Cu_2O$ and $Cu_2(OH)_3Cl$ are the principal corrosion products produced from the Cu5Zn5Al1Sn alloy in marine outdoor conditions, with the

latter largely present at the outmost surface of both Cu and the Cu5Zn5Al1Sn alloy [7,11]. Although a big difference exists in their corrosion rates, the corrosion process is highly administered by the molecular diffusion between the metal surface and the electrolyte [12].

One approach to prevent corrosion of Cu and its alloy is the use of corrosion inhibitors [13,14]. Among the corrosion inhibitors used in practice, cysteine (Cys) is found to be widely employed for Cu protection in various media [15,16]. Cysteine is an amino acid that contains a –SH group in addition to the amino group; this mercaptan group is strongly attracted to copper. A. A. Nazeer et al. have studied the inhibitive effect of cysteine on a Cu10Ni alloy in sulfide containing atmospheres [16]. Based on this study, cysteine can act as a mixed-type inhibitor and cysteine molecules are adsorbed on the alloy surface. The inhibition mechanism of cysteine with copper-based materials is characterized by the formation of a stable Cu (I)–cysteine complex [17]. As confirmed by many researchers, the better corrosion inhibition effect of cysteine relies on its surface adsorption through sulfur atoms [18,19]. I. Milošev et al. on the other hand have studied different amino acids as inhibitors for copper in acidic environments and confirmed that cysteine has higher efficiency. Based on their molecular dynamic simulation results, it has been suggested that the –SH group is in charge of the good protective effect of cysteine [20]. G. M. Abd El-Hafez et al. have also investigated the protective action of methionine, N-acetyl cysteine, and cysteine for a Cu-10Al-5Ni alloy in 3.5 wt.% NaCl solution and they concluded that cysteine showed higher efficiency ascribed to the existence of the mercaptan moiety [21]. On the other hand, our previous study regarding the effect of cysteine on copper metal in corrosive environments also showed its effectiveness on corrosion inhibition.

It can be seen from the above investigated results cysteine can be a very promising corrosion inhibitor for copper and its alloys, and no information has been found on the action of cysteine for the inhibition of Cu5Zn5Al1Sn alloy corrosion. Furthermore, the interaction between the cysteine molecule and the alloy surface in chloride-containing solution has not yet been investigated. In the present paper, the corrosion and corrosion inhibition of the Cu5Zn5Al1Sn alloy in chloride solutions is investigated. The influence of various cysteine concentrations is also taken into account. The performance of the studied inhibitor is evaluated using various electrochemical and surface analysis techniques. The inhibition efficiency, surface behavior, and adsorption mechanism are clearly analyzed and summarized using the obtained experimental results.

## 2. Materials and Methods

A commercial Cu5Zn5Al1Sn alloy in ther form of a 1 mm thick sheet (89 wt.% Cu, 5 wt.% Zn, 5 wt.% Al, and 1 wt.% Sn, equivalent to 84 at.% Cu, 11 at.% Al, 4.5 at.% Zn, and 0.5 at.% Sn) was obtained from Aurubis (Hamburg, Germany). The specimens were wrapped in epoxy resin, exposing a 1 cm$^2$ surface area. Before any test, the alloy was ground by silicon carbide paper from #800 to #2000 and then successively polished with diamond paste to 2.5, 1.5, and 0.5 μm, respectively. The diamond polished specimens were rinsed with analytical grade ethanol and dried in air.

3.5 wt.% sodium chloride (NaCl) was used as the corrosive solution. It was made from an analytical grade reagent of NaCl and deionized water. Cysteine (KC90277-100gm, Suzhou tianke Co. Ltd., Suzhou, China) was used as the inhibitor and stock solution of $1 \times 10^{-2}$ M was prepared by mixing a suitable amount of cysteine in distilled water. Solutions with cysteine concentrations of $1 \times 10^{-3}$ to $1 \times 10^{-5}$ M were set from the stock solution using a dilution method.

The electrochemical cell used was a three-electrode glass cell with a capacity of 400 mL. A silver/silver chloride electrode (Ag/AgCl) was used as the reference electrode. All potential values in the manuscript were referred to using Ag/AgCl. A flat platinum sheet (4 cm$^2$) was used as the counter electrode. An electrochemical workstation Gamry (Reference 600) with a high input resistance of $10^{14}$ Ω, a current detection limit of 60 pA, and a current resolution of 20 aA was used to conduct electrochemical measurements. Prior to the tests of electrochemical impedance spectroscopy (EIS), the specimen was deepened in the test solution for 60 min. EIS was carried out under potentiostatic conditions in the frequency range of 100 kHz to 0.01 Hz at open circuit potential (OCP) with an amplitude of 5 mV.

The EIS data was analyzed by Zsimpwin software to determine a simulated circuit and respective parameters. Tafel curves were obtained from potentiodynamic polarization study which was performed by varying the electrode potential automatically from −800 mV to +400 mV with respect to OCP at a rate of 1 mV/s.

An FEI Quanta 250 scanning electron microscope was used to observe the surface morphology of the Cu5Zn5Al1Sn alloy specimen with and without the cysteine inhibitor in the 3.5 wt.% NaCl solution. X-ray photoelectron spectroscopy (XPS) and Auger electron spectroscopy (AES) measurements were performed on an ESCALAB 250Xi spectrometer with Al Kα X-ray (1486.6 eV) irradiation as the photo source. Binding energy (BE) was calibrated against the C1s line of the aliphatic carbon contamination set at 284.8 eV. Xpspeak41 software was used for data fitting, which was carried out graphically within the constraints of Gaussian peak shapes.

## 3. Results

### 3.1. Potentiodynamic Polarization

Figure 1 presents Tafel polarization curves for the Cu5Zn5Al1Sn alloy exposed to 3.5 wt.% NaCl solution with and without the Cys inhibitor. The trend of polarization curves suggests that a low amount addition of cysteine ($10^{-4}$ M, $10^{-5}$ M) in 3.5 wt.% NaCl solution moves the curves to the lower current region in the anodic potential route compared with the blank solution. Nevertheless, the higher concentration ($10^{-3}$ M, $10^{-2}$ M) of the inhibitor in solution causes a shift of the curves to a more cathodic potential region and decreases the corrosion current meaningfully. Consequently, cysteine influences both the cathodic and anodic reactions. Hence, cysteine is shown to act as a mixed-type inhibitor to the Cu5Zn5Al1Sn alloy in 3.5 wt.% NaCl and to retard majorly cathodic corrosion reactions [22–25]. The corrosion current density (Icorr) and corrosion potential (Ecorr) were calculated by fitting the experimental results with Gamry Echem Analyst software 5.61 and listed in Table 1. The inhibition efficiency (μ%) was also obtained by Equation (1) using parameters listed in Table 1 [26].

$$\mu\% = \frac{\text{Icorr} - \text{Icorr(inh)}}{\text{Icorr}} \times 100\% \tag{1}$$

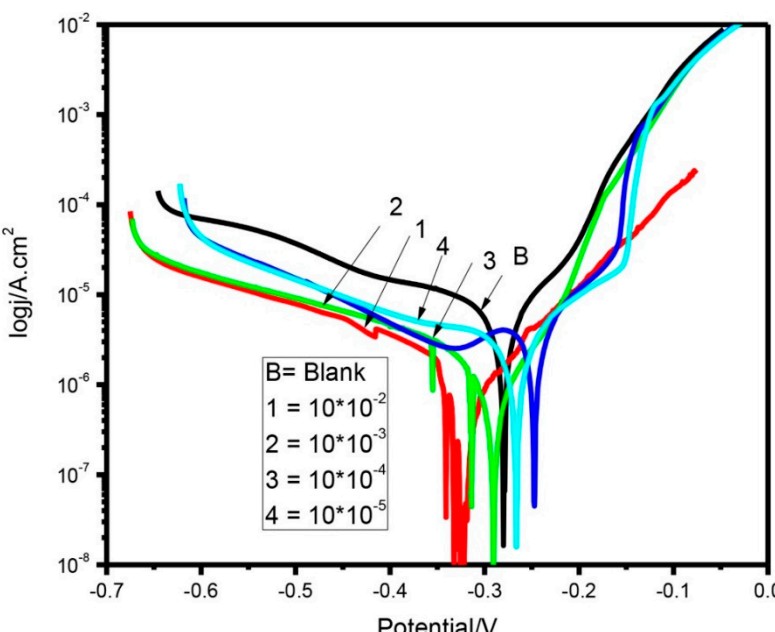

**Figure 1.** Potentiodynamic polarization curves for the Cu5Zn5Al1Sn alloy after 60 min immersion in 3.5 wt.% NaCl in the absence (Blank) and presence of different concentrations of cysteine ($10^{-5}$ to $10^{-2}$).



**Table 1.** Polarization parameters for the Cu5Zn5Al1Sn alloy measured in 3.5 wt.% NaCl solution in the absence and presence of different concentrations of cysteine.

| Cysteine Concentration | $E_{corr}$ (V versus Ag/AgCl) | βa (mV/Decade) | βc (mV/Decade) | $I_{corr}$ ($\mu Acm^{-2}$) | μ% |
|---|---|---|---|---|---|
| Blank (0) | −0.280 | 69.18 | 102.3 | 3.44 | - |
| $10^{-5}$ M | −0.226 | 77.77 | 127.2 | 1.84 | 46.5 |
| $10^{-4}$ M | −0.247 | 48.5 | 63.86 | 1.78 | 48.2 |
| $10^{-3}$ M | −0.292 | 61.23 | 56.96 | 0.483 | 85.9 |
| $10^{-2}$ M | −0.326 | 23.71 | 19.49 | 0.094 | 97.2 |

In this equation Icorr (inh) and Icorr are the current densities of the working electrode in the 3.5 wt.% NaCl solution with and without cysteine, respectively.

The Icorr is inversely related to the inhibitor concentration, i.e., an increase in inhibitor concentration results in a lower value of Icorr. This is probably due to the improved protection of the Cu5Zn5Al1Sn alloy by cysteine molecules. Therefore, the mitigation of the Cu5Zn5Al1Sn alloy deterioration in 3.5 wt.% NaCl solution may associated with the development of a protective layer from the adsorbed inhibitor species on the alloy surface [27]. The adsorption layer of cysteine molecules on the alloy surface can hinder the movement of corrosive species, resulting in a reduction in corrosion rate [2,23,24,28,29].

It is evident from Table 1 that both βc and βa are varied on the addition of the inhibitor. From this behavior, it is suggested that both anodic and cathodic reactions are retarded by the protective layer formed by inhibitor molecules and the alloy/electrolyte interface [3,23,29]. In addition, βc values are greater than βa values at all inhibitor concentrations and the variation in βc with the inhibitor concentration is greater than the variations in βa. Both the results give the same evidence that cysteine is more efficient in inhibition of cathodic reactions than that of anodic reactions [3,30]. The efficiency of cysteine increases with its concentration and reaches a maximum value of 97.2% at the concentration $10^{-2}$ M.

*3.2. Electrochemical Impedance Spectroscopy*

Nyquist and Bode diagrams of the Cu5Zn5Al1Sn alloy in 3.5 wt.% NaCl solution with and without different amounts of cysteine are illustrated in Figure 2a,b, correspondingly. From Figure 2a, the Nyquist diagram for Cu5Zn5Al1Sn alloy in 3.5 wt.% NaCl solution contains a capacitive loop at intermediate frequency and a straight line in the low frequency region. A straight line at low frequency area may be attributed to the presence of the Walberg constant as a result of the movement of soluble metal species from the alloy surface to the bulk solution [23]. The diameter of the capacitive loop rises in the presence of Cys and the straight line in the low frequency region is removed at higher Cys concentrations, which may be attributed to the inhibitive effect of Cys on the corrosion process [31]. The results of Bode plots (Figure 2b) show that the frequency range with maximum phase becomes larger with increasing Cys concentration. There is a shift in the phase maximum to a lower frequency region and the phase angle is raised to about 80°, in the inhibitor concentration of 10−2 M. These results suggest that the protective ability of the Cu5Zn5Al1Sn alloy is increased with inhibitor concentration, showing that inhibitor particles effectively adsorb on the alloy surface [32,33].

The results of the EIS experiment were fitted with a simulated circuit (Figure 3) and the value of electrical elements that make up the equivalent circuit are presented in Table 2. The equivalent circuit parameters are solution resistance (Rs), film resistance (Rf), film capacitance (Qf), charge transfer resistance (Rct), double layer capacitance (Qdl), and Warburg resistance (W), which result from ionic diffusion of the corrosion product to the electrode surface. The EIS fitting result of the Cu5Zn5Al1Sn alloy in blank 3.5%wt NaCl solution shows a Warburg resistance and a lower value of Rct, as shown in Figure 2a and Table 2, respectively. The Warburg impedance is included to account for the diffusion of corrosion products to the bulk solution and/or dissolved oxygen to the electrode surface. With a lower

Cys concentration, the Warburg impedance remains unchanged, and Rf and Qf are observed to address the film formed between the Cys molecules and alloying elements. The presence of W is attributed to the low inhibition efficiency of the lower concentration of Cys, which may be related to the thinner and unstable film formed which can be easily dissolved to the bulk solution. When more than $10^{-4}$ M Cys is added to the solution, the Warburg impedances vanish and the Rct values become larger. This shows the presence of high resistance to transfer charge from bulk solution to specimen surface and/or and specimen surface to bulk solution; hence, the diffusion process is controlled. This may be attributed to the development of a thick and protective Cys/metal film on the alloy surface.

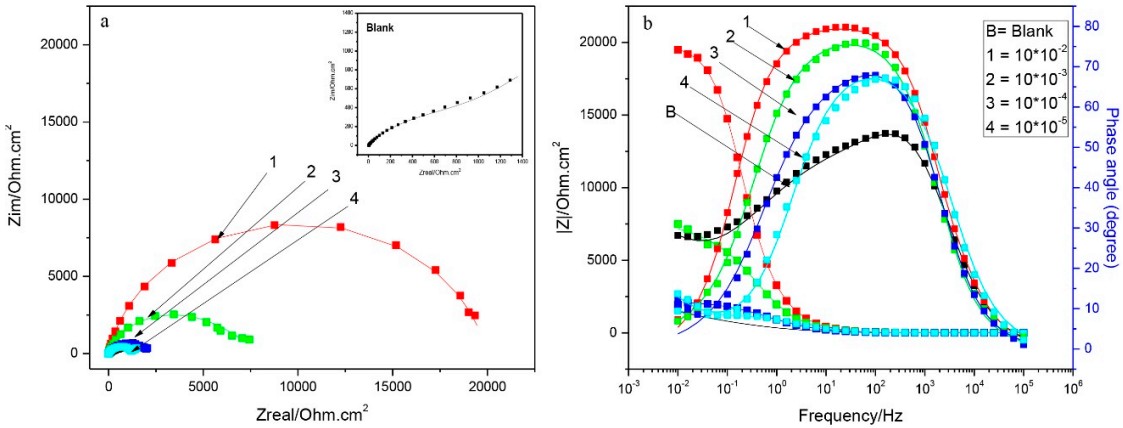

**Figure 2.** Nyquist (**a**) and Bode (**b**) plots for the Cu5Zn5Al1Sn alloy in 3.5 wt.% NaCl with and without different concentrations of cysteine ($10^{-5}$ to $10^{-2}$) (scattered: experimental data; lines: fitting data).

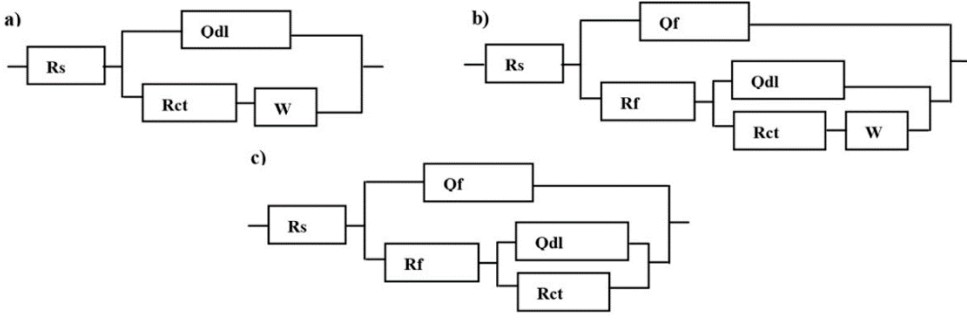

**Figure 3.** Electrical equivalence circuit for Cu5Zn5Al1Sn alloy in 3.5 wt.% NaCl without R(Q(RW)) (**a**), with $10^{-5}$ M Cys, R(Q(R(Q(RW)))) (**b**) and $10^{-4}$–$10^{-2}$ M Cys, R(Q(R(QR))) (**c**). Legend: solution resistance (Rs), film resistance (Rf), film capacitance (Qf), charge transfer resistance (Rct), double layer capacitance (Qdl), and Warburg resistance (W).

**Table 2.** Impedance parameters for the Cu5Zn5Al1Sn alloy in 3.5 wt.% NaCl with and without different concentrations of cysteine obtained by using ZSimpWin 3.50.

| Cys Conc. (M) | Rs (Ω) | Qf | Rf (Ω) | Qct | Rct (KΩ) | W | Rp (KΩ) | µ% |
|---|---|---|---|---|---|---|---|---|
| Blank | 2.668 | | | $1.55 \times 10^{-4}$ | 692.7 | 0.0032 | 0.695 | - |
| $10^{-5}$ | 3.224 | $8.09 \times 10^{-5}$ | 588.1 | $1.64 \times 10^{-4}$ | 1169 | 0.0053 | 1.197 | 40.76 |
| $10^{-4}$ | 2.894 | $1.05 \times 10^{-4}$ | 212.7 | $1.61 \times 10^{-4}$ | 1.986 | | 2.201 | 67.78 |
| $10^{-3}$ | 3.345 | $8.24 \times 10^{-5}$ | 2382 | $1.10 \times 10^{-4}$ | 6.399 | | 8.784 | 91.92 |
| $10^{-2}$ | 3.067 | $5.28 \times 10^{-5}$ | 12.02 | $4.41 \times 10^{-6}$ | 19.940 | | 19.955 | 96.44 |

Inhibition efficiency was obtained by using Equation (2) [34–36], the values of which are listed in Table 2. The inhibition efficiency of Cys for Cu5Zn5Al1Sn alloy corrosion in 3.5 wt.% solution is seen to increases with its concentration, and a maximum inhibition efficiency value of 96.44% is recorded for the case of the Cys concentration being $10^{-2}$ M. The inhibition efficiencies obtained from polarization

and EIS investigations are similar and follow the same trend, which implies that the explanation for Cu5Zn5Al1Sn alloy corrosion characteristics on the basis of Icorr and Rp could be well-thought-out as valid and reliable.

$$\mu\% = \frac{Rp(inh) - Rp}{Rp(inh)} \times 100\% \tag{2}$$

where μ, Rp, and Rp (inh) are inhibition efficiency, and polarization resistance for uninhibited and inhibited surfaces, respectively.

The inhibition efficiency results obtained from potentiodynamic (PD) and EIS experiments were used to observe the adsorption behavior of cysteine on the Cu5Z5Al1Sn alloy surface. Therefore, different adsorption isotherms were tested to determine the best suitable model to explain surface phenomena [23,37,38].

$$\frac{C}{\theta} = \frac{1}{Kads} + C \tag{3}$$

$$\theta = \frac{RT}{b}lnKads + \frac{RT}{b}lnC \tag{4}$$

In this equation, C, θ, $K_{ads}$, and b are inhibitor concentration, the inhibitor's surface coverage, the equilibrium constant related to the interfacial molecular adsorption-desorption phenomena, and a constant relevant to the properties of both adsorbent and adsorbate in the Temkin model, respectively.

Figure 4 shows plots of Langmuir and Temkin adsorption isotherms determined by the respective equations listed in Equations (3) and (4), respectively. As observed in Figure 4a, a slope of 0.99 with $R^2 = 0.99$ derived from EIS and PD results which is close to unity can be observed from a C/θ versus C linear plot for the case of the Langmuir adsorption isotherm. On the other hand, a regression of 0.89 and 0.83 acquired from EIS and PDP, respectively, was obtained from the plot of θ versus ln C based on the Temkin adsorption isotherm (Figure 4b). By comparing the two isotherms, the Langmuir adsorption isotherm, which was seen to have less fitting error, was chosen as the more suitable model to describe the surface inhibition mechanism of cysteine for Cu5Z5Al1Sn alloy corrosion.

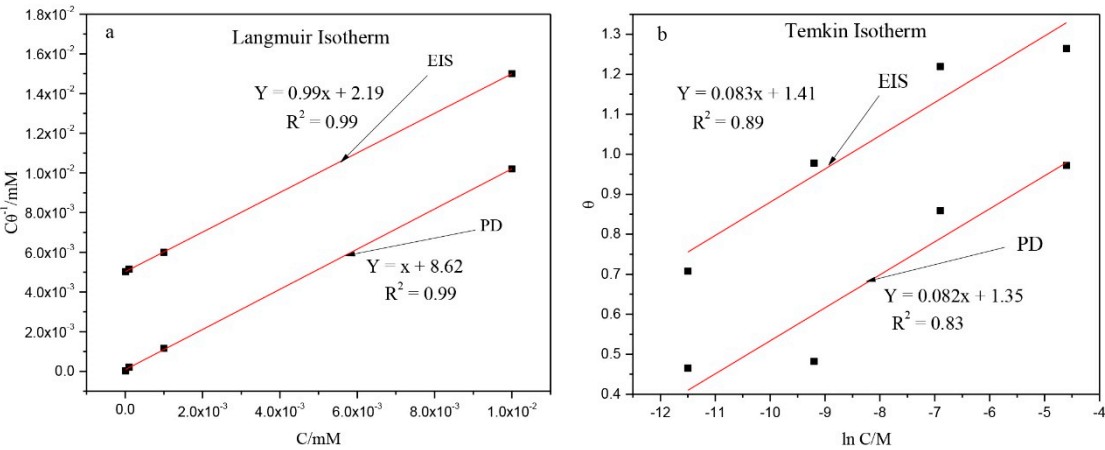

**Figure 4.** Langmuir (**a**) and Tamkin (**b**) adsorption plots of Cu5Z5Al1Sn alloy in 3.5 wt.% NaCl in the presence of the addition of different concentrations of cysteine obtained by EIS and PD experiments.

### 3.3. SEM Analysis

Further analysis was performed to examine the surface morphology and elemental characterizations and their effects on corrosion behavior. Figure 5 illustrates the SEM images of the Cu5Zn5Al1Sn alloy exposed to 3.5 wt.% NaCl solution without and with different amounts of Cys. However, the uninhibited copper alloy surface (Figure 5b) is highly corroded and becomes rough as a result of the aggressive attack the from corroding solution. A very different surface morphology is observed in the presence

of $10^{-4}$ M Cys as, shown in Figure 5c. The inhibitor molecules, which partly cover the surface, are seen to be formed. By contrast, Figure 5d does not show any corrosion attack and has almost the same morphology as the unexposed surface, suggesting that the addition of $10^{-2}$ M Cys leads to a more inhibited corrosion, the surface of which is almost the same as an unexposed polished one (Figure 5a). Thus, it is obvious that corrosive attack is considerably restricted by $10^{-2}$ M Cys. It is assumed that inhibitor molecules adsorb on the alloy surface and a smoother surface forms compared to the surface treated with the blank 3.5 wt.% NaCl solution.

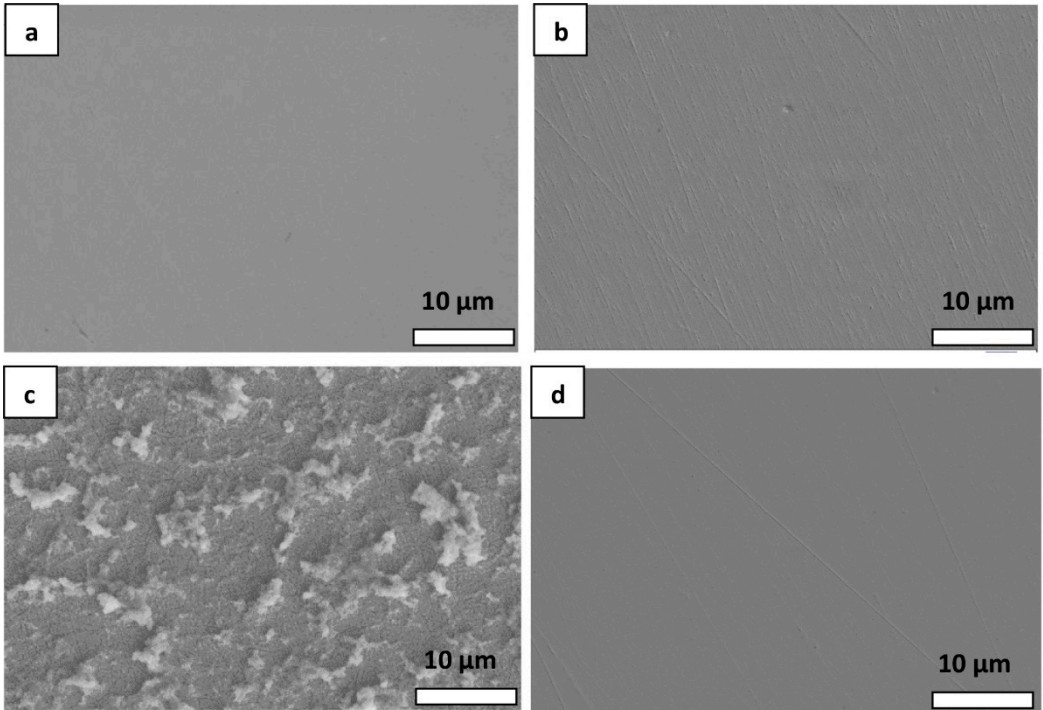

**Figure 5.** SEM images of Cu-5Zn-5Al-1Sn alloy before (**a**) and after immersion in 3.5 wt.% NaCl, without (**b**) and with $10^{-4}$ M of cysteine (**c**) and $10^{-2}$ M cysteine (**d**), respectively.

### 3.4. XPS and Auger Results

In order to further understand the surface composition of the uninhibited and inhibited copper alloy, high-resolution XPS Cu2p and Auger CuLM2 spectra of the Cu5Zn5Al1Sn alloy surface without and with the addition of various concentrations of cysteine were recorded and are shown in Figure 6. The Cu 2p profiles of Cu(0) and Cu(I) are similar, and as a result it is difficult to use the XPS Cu 2p spectrum alone to differentiate them, whereas the binding energy of the Auger peak of Cu(I) is about 2 eV higher than the Auger peak of Cu(0), meaning the CuLM2 spectrum is necessary to further investigate the valence state of copper. In all cases, there are two deconvoluted peaks of Cu2p 3/2 at different binding energies (Figure 6a). The interpretation of the peaks is different as the reaction taking place on the alloy surface is not same in the blank and Cys-inhibited solutions. In the blank solution, the corrosion products may exist as the form of oxides by reactions with the dissolved oxygen. While in the presence of inhibitors, the inhibitor particles have the opportunity to participate in surface reaction. In blank 3.5 wt.% NaCl solution, the deconvoluted Cu2p peaks at 932.99 and 934.83 eV (Figure 6a) may be assigned to Cu/$Cu^+$ and $Cu^{2+}$, respectively. In addition to the XPS result, the Auger spectrum obtained in the blank solution shows an Auger peak at 570.11 eV, which is a typical characteristic of cuprous ion ($Cu^+$), with a band broadening to the lower binding energy. The band broadening is an indication of the presence of metallic copper, which may be attributed to the uncovered alloy surface. Generally, the XPS and Auger results of the blank solution show the presence of Cu, $Cu^+$, and $Cu^{2+}$ species on the alloy/solution interface, which may be attributed to the uncovered alloy surface.

With regard to the Cu2p 3/2 obtained after immersion in solution with $10^{-4}$ M Cys, the deconvoluted peaks are located at 932.53 and 934.41 eV, respectively. Similarly, when the surface is inhibited with $10^{-2}$ M Cys, the Cu2p 3/2 peak is deconvoluted into peaks located at 932.62 and 934.10 eV. The peaks may be assigned to Cu/Cu$^+$ and Cu$^{2+}$ ions, respectively, in both cases [39–42]. In the solution containing different concentrations of cysteine ($10^{-4}$ and $10^{-2}$ M), the Auger peaks obtained at 571.05 and 571.02 eV may both be assigned to Cu$^+$. The binding energy of the Cu$^+$ peak on the Auger spectra increases in the presence of Cys, which may be associated with the formation of a film containing organic inhibitor molecules typically centered at higher binding energies than the binding energy for the oxide species [43–47]. Both in the absence and presence of Cys, the Cu2p XPS spectra demonstrate the Cu (II) satellite peaks attributed to the presence of copper (II). Furthermore, there is no Auger peak related to Cu$^{2+}$ other than a band broadening in the blank solution, which may be ascribed to the lower content of the Cu$^{2+}$ species. The uninhibited alloy surface and that of the inhibited one with lower Cys concentration ($10^{-4}$ M) have a higher peak intensity, which could be attributed to the poor coverage area of corrosion products and the metal/inhibitor film, respectively [48].

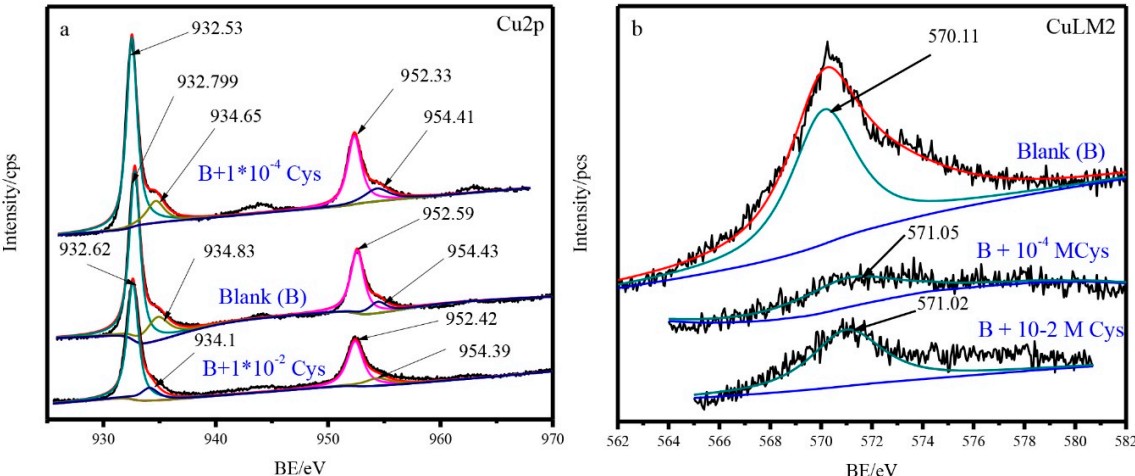

**Figure 6.** High-resolution XPS Cu2p 3/2 spectra (**a**) and Auger spectra (**b**) of the Cu5Zn5Al1Sn alloy surface without and with addition of inhibitors in 3.5 wt.% NaCl. Legend: BE, binding energy.

High resolution XPS Zn2p spectra (Figure 7a) show two peaks for the alloy surface, both without and with different concentrations of Cys, which may be assigned to Zn2p 3/2 and Zn2p 1/2. The Zn2p 3/2 peak at 1021.75 and 1021.76 eV in the blank and $10^{-4}$ Cys inhibited surfaces may be assigned to Zn$^{2+}$, which is associated with the presence of corrosion product [49–51]. However, the peak observed on the surface inhibited with $10^{-2}$ M Cys is almost negligible due to low Zn$^{2+}$ species on the alloy surface. This phenomenon indicates that the alloy surface is not fully protected by inhibitor molecules in the presence of lower cysteine concentration. Obviously, the dezincification rate of the Cu5Zn5Sn1Al alloy is strongly bounded by the surface film/inhibitor film of the alloy surface. A more dense/integrated surface film leads to a lower dezincification rate. It can also be deduced that the formed inhibited film may rise from other alloying metals of the Cu5Zn5Sn1Al alloy, which will probably be a copper-cysteine film.

The Sn3d peak may be deconvoluted into two different peaks: Sn5/2 and Sn3/2. The peaks centered at 486.65 eV, 486.77 eV, and 486.58 eV correspond to the Sn5/2 peaks, which were obtained in blank solution, with $10^{-4}$ M Cys and with $10^{-2}$ M Cys, respectively (Figure 7b). In all cases, the peak values correspond to ionic tin (Sn$^{4+}$) [39,52–56], which may indicate the formation of the compound of Sn on the alloy surface, meaning, therefore, the contribution of Sn$^{2+}$ to the surface film should not be ignored. In addition to the main Sn3d peaks, satellite peaks at higher binding energies of 499.05 and 499.08 eV appear for the blank and the surface-inhibited surface with lower Cys concentration ($10^{-4}$ M), respectively, which may be attributed to the presence of more tin ion contents on the alloy surface [57].

In the meantime, the atomic ratio shown in Table 3 reveals that the inhibitor free alloy surface contains a greater amount of tin. On the other hand, the alloy surface inhibited with $10^{-2}$ M Cys contains an almost insignificant amount of $SnO_2$, which may result in the development of the inhibitor film and prevent the formation of the corresponding metal oxide.

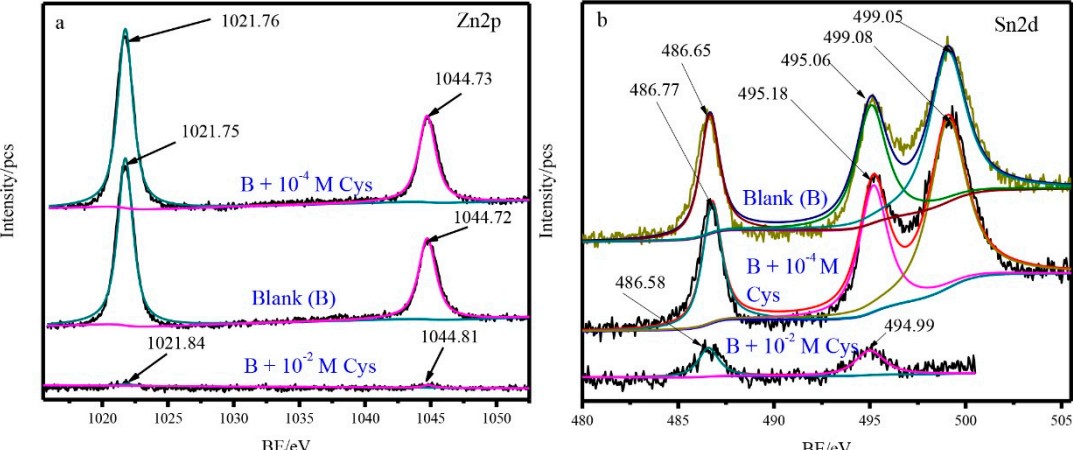

**Figure 7.** High-resolution XPS of Zn2p (**a**) and Sn3d (**b**) spectra of Cu5Zn5Al1Sn alloy surface without and with addition of inhibitors in 3.5 wt.% NaCl solution.

**Table 3.** Composition (atomic %) of the surface of Cu5Zn5Al1Sn alloy samples before and after the addition of inhibitors in a 3.5 wt.% NaCl solution.

| Element | Atomic % | | |
|---|---|---|---|
| | **Blank** | **Blank + $10^{-4}$ M Cys** | **Blank + $10^{-2}$ M Cys** |
| C1s | 37.9 | 33.54 | 49.4 |
| N1s | 2.77 | 4.52 | 7.78 |
| O1s | 36.99 | 37.07 | 29.38 |
| S2p | 0 | 2.1 | 3.29 |
| Cu2p | 13.22 | 14.42 | 9.36 |
| Zn2p | 7.98 | 7.26 | 0.63 |
| Sn3d | 1.14 | 1.08 | 0.16 |

The deconvoluted C1s spectrum illustrated in Figure 8a for the golden alloy in the absence of cysteine may be attributed to three peaks which indicate three chemical forms of the C element present on the alloy surface. The three peaks described above are located at 284.75, 285.73, and 288.59 eV. The component at 284.6 eV is assigned to the non-oxidized carbon containing (C–C) composition. Moreover, the features at BE values of 285.73 eV and 288.59 eV can be assigned to the groups containing a carbon–oxygen bond, i.e., a C–O group like ether or hydroxyl, and adventitious carbon which is usually found on the metal surfaces and results from adsorbed oxidized carbonaceous species from the atmosphere, respectively [44,45,58]. For the case involving the presence of $10^{-4}$ M cysteine, the C1s spectrum (Figure 7a) also shows three peaks at 288.54, 286.38, and 284.88 eV, and the three deconvoluted peaks for the case of $10^{-2}$ M Cys are located at 288.68, 286.21, and 284.66 eV, which match with the COO−, C−$NH_2$, and C− SH groups, respectively [39,40]. The increment of the atomic ratio of carbon (Table 3) of the $10^{-2}$ M Cys inhibited surface can be ascribed to the existence of the high amount of cysteine on the alloy surface.

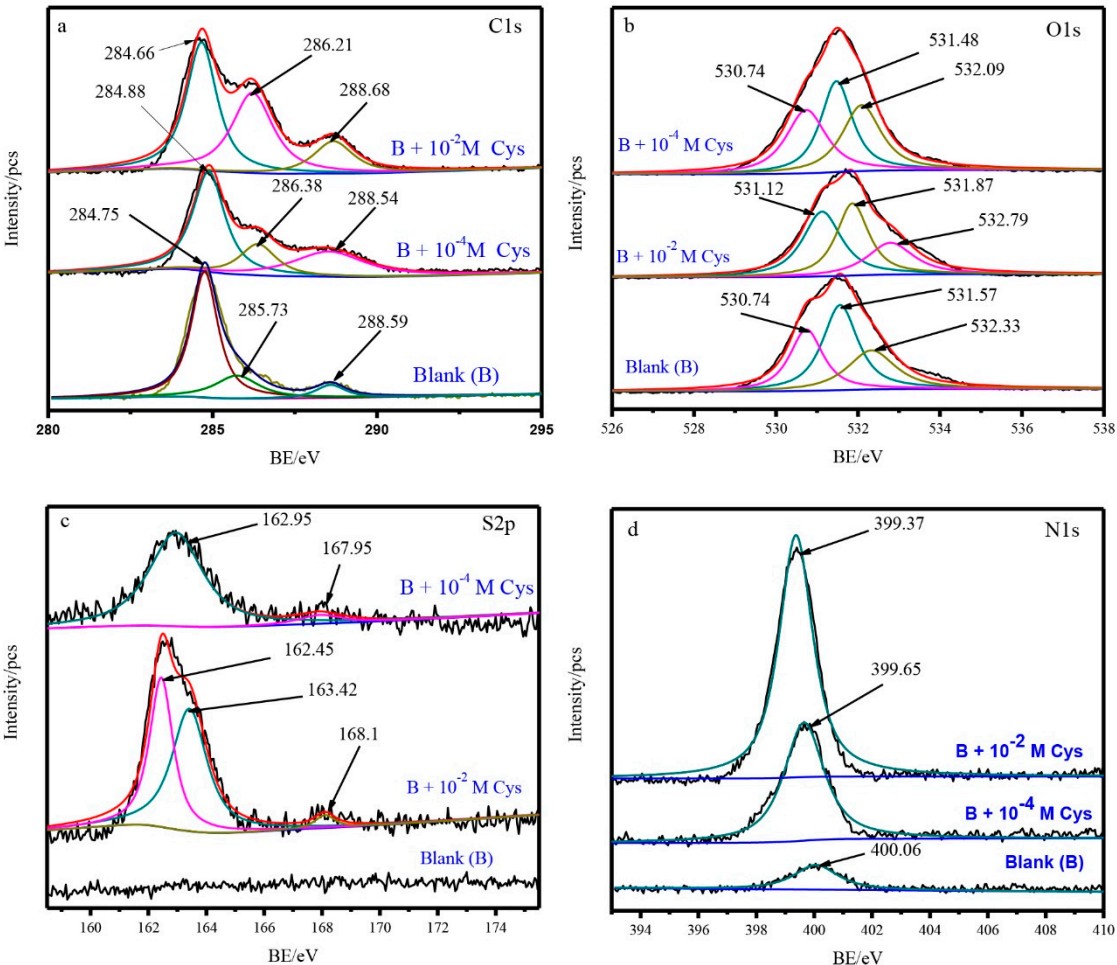

**Figure 8.** High-resolution XPS C1s (**a**), O1s (**b**), S2p, and N1s spectra of the Cu5Zn5Al1Sn alloy surface before and after addition of inhibitors in 3.5 wt.% NaCl solution (**c,d**).

The deconvoluted O1s spectra of the Cu5Zn5Al1Sn alloy in the 3.5 wt.% NaCl solution without and with the Cys inhibitor are shown in Figure 8b. There are three distinct peaks for the blank alloy surface in 3.5 wt.% NaCl solution, with the first peak located at 530.74 eV being attributed to $O^{2-}$, which could be related to oxygen atoms bound to the constituent metal oxides [48,59,60]. Since the major corrosion product on the outermost part of the alloy is derived from copper, the dominant metal oxide relies on copper [8], whereas the contribution of other constituent metals (Zn, Sn, and Al) cannot be ignored. The presence of Zn and Sn oxides are confirmed by their respective XPS spectrum. On the other hand, it is impossible to obtain peak analysis for Al (neither Al2p nor Al2s can be used) because of the peak overlapping with Cu3p and Cu3s [8]. The second peak observed at 531.57 eV may be ascribed to $OH^-$, which can be associated with the occurrence of hydrous copper, zinc, tin, and aluminum oxides.

The third peak at 532.33 eV may be assigned to atmospheric oxygen, which is similar to the carbon contamination [42]. With the addition of $10^{-4}$ M of Cys, the O1s peak is deconvoluted into three peaks located at 530.74, 531.48, and 532.09 eV, which could be assigned to metal oxides, hydrous metal oxides, and C=O or –CON arising from cysteine molecules, respectively. The presence of metal oxide can be ascribed to partial coverage of the metal surface by the inhibitor film [42,61,62]. The O1s spectrum of the sample containing $10^{-2}$ M Cys may be deconvoluted into three peaks at 531.12, 531.87, and 532.79 eV, which can be attributed to C-O from carboxyl and C=O or -CON, respectively [39]. In contrast to the uninhibited and the $10^{-4}$ M Cys inhibited alloy surface, the alloy inhibited with the high Cys ($10^{-2}$ M) concentration shows no peak associated with the presence of metal oxide. It is

indicated that the alloy surface is fully covered by the inhibitor film after the addition of $10^{-2}$ M cysteine [63].

Figure 8c shows the deconvoluted S2p spectrum of Cu5Zn5Al1Sn alloy in the absence and presence of cysteine. There is no peak found on the S2p spectrum of the blank Cu5Zn5Al1Sn alloy in 3.5 wt.% NaCl attributed to the nonappearance of any sulfur species. In the presence of $10^{-4}$ M Cys, two peaks located at 162.95 and 167.95 eV can be recognized, and for the case of $10^{-2}$ M Cys, the three deconvoluted peaks are located at 162.45, 163.42, and 168.1 eV. The lower binding energies (162.95 and 162.45) may be assigned to the metal–sulfur interaction and the upper ones (167.95 and 168.10) are ascribed to the -SH group of the cysteine molecule [42,61,63,64]. The deconvoluted peaks at 162.45 and 163.42 eV obtained from the $10^{-2}$ M Cys inhibited surface result from band broadening and because of the presence of more sulfur content they are considered to be peaks derived from the same component [43,65]. The intensity of S2p peaks and percentage ratio of S (Table 3) increases with cysteine concentration, suggesting that the efficiency of cysteine is concentration-dependent and more Cys molecules are adsorbed on the Cu5Zn5Al1Sn alloy surface at the higher inhibitor concentration.

The N1s spectrum for the uninhibited/inhibited surface has only one deconvoluted peak, as shown in Figure 8d, for all cases. The peak at 400.02 eV obtained from the blank solution can be assigned to N raised from the atmosphere during sample preparation. The peaks at 399.65 and 399.37 eV are obtained from $10^{-4}$ M and $10^{-2}$ M, respectively, can be assigned to the secondary nitrogen (–NH) attributed to the presence of organic matrix belonging to cysteine [39]. The peak beyond 400 eV in the N1s spectra of the inhibited surfaces is absent, which shows that there is no -$N^+H^-$ group [66]. Therefore, the above phenomenon indicates that the adsorption site of cysteine on the surface of the studied alloy does not rely on the N group. An atomic ratio of nitrogen (Table 3) in a surface inhibited with $10^{-2}$ M Cys confirms that the greater amount of inhibitor species are adsorbed on the alloy surface, and hence a better protection effect is displayed.

## 4. Discussion

The corrosion characteristics of the Cu5Zn5Al1Sn alloy in 3.5 wt.% NaCl solution were examined by different electrochemical and surface analysis techniques. It has been proven from electrochemical tests that the corrosion of the Cu5Zn5Al1Sn alloy in 3.5 wt.% NaCl solution decreases on the addition of the cysteine inhibitor ( Figure 1; Figure 2). The effect of cysteine concentration was also investigated and the highest inhibition efficiencies of 97.2% and 96.44% were obtained at a cysteine concentration of $10^{-2}$ M based on potentiodynamic and EIS investigations, respectively. Morphological analysis based on SEM shows a better protected alloy surface in the presence of the highest concentration ($10^{-2}$ M) of cysteine. Furthermore, composition and the inhibition mechanism were further determined by carrying out elemental analysis using XPS techniques.

From potentiodynamic testing it was determined that the current density decreases upon addition of the inhibitor, which can be accredited to the inhibitive effect of cysteine. In addition, the corrosion potential was observed to shift to the negative direction for the inhibited samples, with a potential difference between the blank and inhibited samples of less than 85 mV, which can be ascribed to the mixed inhibitive effect of cysteine for the Cu5Zn5Al1Sn alloy in 3.5 wt.% NaCl solution and which has a greater effect to retard cathodic reaction [33,67]. A diversity of adsorption isotherms like Langmuir, Temkin, Freundlich, and Frumkin are often used to model experimental results to understand surface reactions [23,37]. An adsorption model to describe the inhibition mechanism of cysteine of the Cu5Zn5Al1Sn alloy in 3.5 wt.% NaCl with different concentrations of cysteine was generated using the data obtained by PDP and EIS experiments. In this paper, the Langmuir and Tamkin adsorption isotherms (Figure 4) have been used to find out the most appropriate model to describe the inhibition mechanism of cysteine for the copper alloy. The Langmuir adsorption isotherm is proposed as a suitable model to describe the inhibition mechanism of cysteine on the Cu5Zn5Al1Sn alloy surface, which can be accredited to the adsorption of inhibitor particles on the active site of the Cu5Zn5Al1Sn alloy substrate [68,69].

In the blank solution, the appearance of Warburg impedance in the simulated circuit of the EIS result (Figure 3) reflects the diffusion process of the ionized alloy molecules from the alloy surface to the bulk solution, or the diffusion process of dissolved oxygen from the wholesale solution to the superficial of the alloy [70,71]. Composition analysis does help to study the corrosion phenomena of the Cu5Zn5Al1Sn alloy in 3.5 wt.% NaCl solution. A Cu2p 1/2 peak at 932.79 eV and Auger CuLMM peak at 570.11 eV with band broadening to low BE direction are observed and may be attributed to the presence of both metallic and ionic copper as corrosion products. There are other alloying elements present on the alloy surface which illustrate their involvement in the corrosion process. As can be seen from Table 3, the ionic ratio of Zn and Sn ions in the blank solution are higher than that for the inhibited surface. This can be ascribed to the ionization of Zn and Sn species in blank solution; they become bounded by the inhibiter film after the addition of cysteine. The damaged surface can also be observed from the SEM image (Figure 5b), which strengthens the results of the electrochemical and XPS experiments.

A relatively protected surface appears after the addition of a small amount of cysteine. The electrochemical experiments (PDP and EIS) show the inhibitive effect of cysteine (Figures 1 and 2), which is confirmed by the decrease in current density and an increase in impedance in the potentiodaynamic polarization and EIS curves, respectively. On the other hand, the inhibition efficiency obtained by electrochemical parameters of both potentiodynamic polarization and EIS is seen to increase with the increase in inhibitor concentration. There are S and N constituents in the inhibited surfaces, indicating the presence of the inhibitor film. However, the atomic ratio of this species is lower compared to a surface which has a high cysteine concentration. As for the alloying elements, the atomic ratio of Cu2p increases and the atomic ratio of Zn and Sn decreases in the presence of $10^{-4}$ M cysteine, inferring the bounding of the alloy surface by the Cu-inhibitor film.

The atomic ratio of alloying elements other than Cu decreases with the addition of cysteine and becomes negligible at higher cysteine concentration ($10^{-2}$ M), which shows that the inhibitor film is mainly formed by the dominant alloy constituent (Cu) [8] and the inhibitor. In addition, the surface inhibited with $10^{-2}$ M cysteine concentration shows an almost similar morphology, with an unexposed polished surface surmising a good and almost full coverage of the alloy by cysteine molecules by forming a metal/inhibitor film. It is difficult to obtain XPS data for Al due to peak overlap with Cu3s and Cu3p. Schematic representation of the probable corrosion mechanism of Cu5Z5Al1Sn alloy in 3.5 wt.% NaCl solution with and without the Cys inhibitor is illustrated in Figure 9. As mentioned above, the existence of alloying elements on the blank surface can be attributed to the ionization of these metals in 3.5 wt.% NaCl forming the respective metal oxide and hydroxide (Figure 9a) [11]. On the other hand, the removal of some alloying elements, especially Zn and Sn in the presence of low concentration of the inhibitor, indicates the alloy surface is partially protected by an adsorption film between the dominant metal ion (Cu$^+$, Cu$^{2+}$) and inhibitor molecules (Cu(I)-Cys, Cu(I)-Cys-Cu(II)-Cys). With the increase of Cys concentration to $10^{-2}$ M, the main alloying element existing in the surface film is copper (Table 3), and it is believed that the continuous surface inhibited film formed between the copper ion and Cys molecules has existed on the Cu5Zn5Al1Sn alloy surface (Figure 9c). Therefore, better corrosion efficiency is obtained when $10^{-2}$ M Cys is added into the 3.5 wt.% NaCl solution.

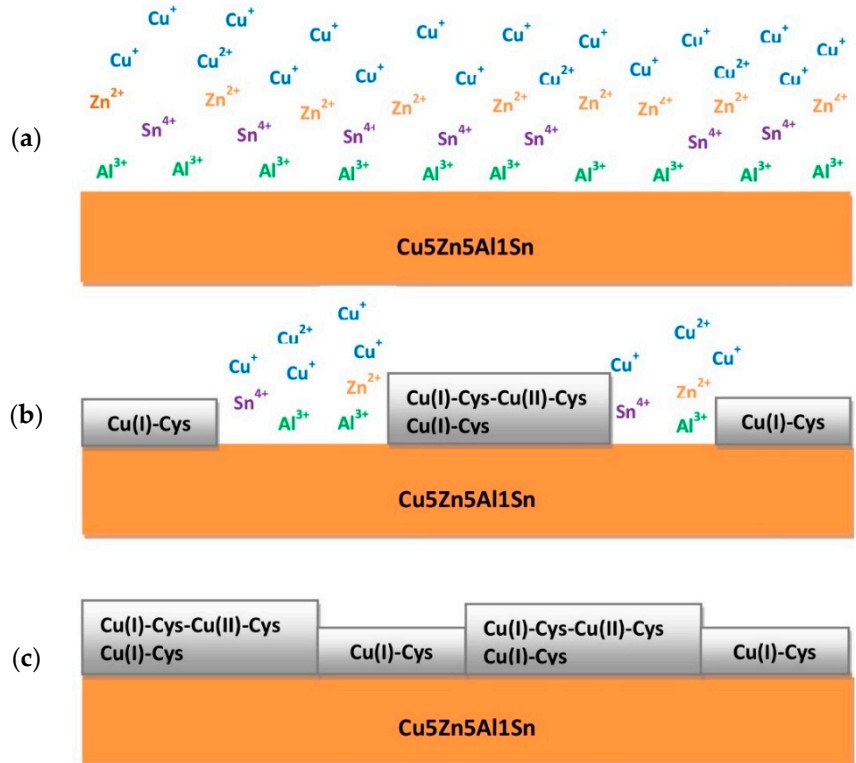

**Figure 9.** Schematic representation of the corrosion mechanism of the Cu5Zn5Al1Sn alloy in 3.5 wt.% NaCl solution before (**a**) and after addition of $10^{-4}$ M (**b**) and $10^{-2}$ M (**c**) cysteine.

## 5. Conclusions

In this work, the corrosion behavior of a Cu5Zn5Al1Sn alloy was examined in a 3.5 wt.% NaCl solution in the absence and presence of cysteine. Potentiodynamic polarization studies showed that cysteine acts as a mixed-type inhibitor during the corrosion process. The inhibition efficiency increased with an increase in the concentration of cysteine and reached 97.2% in the presence of $10^{-2}$ M cysteine. Copper was the main corrosion product formed on the Cu5Zn5Al1Sn alloy in 3.5 wt.% NaCl solution. The outermost surface also contained to a small extent Zn- and Sn-rich constituents, as confirmed by XPS investigation. The content of alloy constituents other than Cu decreased in the presence of cysteine and became negligible at a cysteine concentration of $10^{-2}$ M. The inhibition mechanism relies on the formation of a binding Cu(I)-Cys film and/or Cu(I)-Cys-Cu(II)-Cys film on the alloy surface, which protect the alloy from corrosion attack.

**Author Contributions:** Conceptualization, K.W.S., F.H., and Y.J.; data curation, K.W.S. and F.H.; formal analysis, K.W.S. and F.H.; investigation, K.W.S. and F.H.; methodology, K.W.S., F.H., and Y.X.; project administration, Y.J.; resources, Y.J.; software, K.W.S. and F.H.; supervision, Y.X., L.W., and Y.J.; writing—original draft, K.W.S.; writing—review & editing, F.H., Y.X., and Y.J.

**Funding:** This research was funded by the Development and Reform Committee of PRC, (no. YYXM-1412-0001), Fundamental Research Funds for the Central Universities (no. FRF-TP-16-040A1) and 111 Project, (no. B12012).

**Conflicts of Interest:** The authors declare no conflict of interest. The funders had no role in the design of the study; in the collection, analyses, or interpretation of data; in the writing of the manuscript, or in the decision to publish the results.

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
