# Peer review of "The Protection Role of Cysteine for Cu-5Zn-5Al-1Sn Alloy Corrosion in 3.5 wt.% NaCl Solution"

_applsci, doi:10.3390/app9183896_

Round 1
Reviewer 1 Report
Authors:
Pag. 5, lines 143-145:
"From Figure 2a, the Nyquist diagram for Cu5Zn5Al1Sn alloy in 3.5 wt.% NaCl solution contains a capacitive loop at intermediate frequency and an inductive loop and a straight line at low frequency region"
Reviewer:
Where is the inductive loop?
Authors:
Pag. 5, line 158:
"The equivalent circuit parameters are..."
Reviewer:
These are electrical elements that make up the equivalent circuit. They are not parameters.
Comments
This is an ease paper that could be interesting for the scientific community interested in metallic corrosion, electrochemistry and corrosion protection by environmentally friendly corrosion inhibitors.
It would have been desirable that the effort devoted to the interpretation of impedance diagrams were deeper. The authors would have shown the fit results obtained by applying equivalent circuits of one and two time constants. It is not clear that the use of the equivalent circuit proposed in Figure 3c leads to better fitting results than when using a circuit R (RQ).
It would have been desirable for the authors show and compare the impedance fit plot obtained when these two circuits are used. It would be also desirable to show an analysis of errors and the values obtained for the chi-square parameter by applying both equivalent circuits.
Reviewer 2 Report
Cysteine is widely used for Copper protection in various media, this paper examined the effect of the cysteine as corrosion inhibitor on Cu5Zn5Al1Sn alloy in 3.5 wt.% NaCl solution. A relatively protected surface is appeared after addition of small amount of cysteine. The electrochemical experiments (PDP and EIS) show the inhibitive effect of cysteine. Surface morphology observation confirmed that Cu-5Zn-5Al-1Sn alloy was damaged in 3.5 wt.% NaCl solution and could be inhibited by using cysteine inhibitor. The impact of alloying elements on corrosion mechanism was further examined by the surface analysis techniques.
Author Response
Response to Reviewer 1 Comments
Point 1: Extensive editing of English language and style required.
The authors thank for the effort to review our manuscript. We tried to make an extensive English edition with our friends and made several revisions. All the revisions are highlighted in the manuscript